# Enteric Pathogens Detected in Children under Five Years Old Admitted with Diarrhea in Moshi, Kilimanjaro, Tanzania

**DOI:** 10.3390/pathogens12040618

**Published:** 2023-04-19

**Authors:** Ephrasia A. Hugho, Happiness H. Kumburu, Nelson B. Amani, Bahati Mseche, Athanasia Maro, Lilian E. Ngowi, Yudathadei Kyara, Grace Kinabo, Kate M. Thomas, Eric R. Houpt, Jie Liu, Tine Hald, Blandina T. Mmbaga

**Affiliations:** 1Biotechnology Research Laboratory, Kilimanjaro Clinical Research Institute, Moshi 25102, Tanzania; 2Faculty of Medicine, Kilimanjaro Christian Medical University College, Moshi 25102, Tanzania; 3Department of Pediatrics, Kilimanjaro Christian Medical Centre, Moshi 25102, Tanzania; 4Centre for International Health, Dunedin School of Medicine, University of Otago, Dunedin 9054, New Zealand; 5New Zealand Food Safety, Ministry of Primary Industries, Wellington 6011, New Zealand; 6Division of infectious Diseases and International Health, University of Virginia, Charlottesville, VA 22902, USA; 7School of Public Health, Qingdao University, Qingdao 266073, China; 8Research Group for Genomic Epidemiology, Technical University of Denmark, 2800 Lyngby, Denmark

**Keywords:** diarrhea, diarrheal pathogens, children, Tanzania

## Abstract

Despite the availability and wide coverage of rotavirus vaccinations in Tanzania, there is still a significant number of diarrhea cases being reported, with some patients requiring hospital admission. We investigated diarrhea-causing pathogens and determined the effect of co-infection on clinical symptoms. Total nucleic acid was extracted from archived stool samples (N = 146) collected from children (0–59 months) admitted with diarrhea in health facilities in Moshi, Kilimanjaro. Pathogen detection was performed using the quantitative polymerase chain reaction with custom TaqMan Array cards. The Poisson model was used to determine the effect of co-infection on clinical presentation during admission. Of all the participants, 56.85% were from rural Moshi with a median age of 11.74 months (IQR: 7.41–19.09). Vomiting (88.36%) and a fever (60.27%) were the most frequent clinical manifestations. At least one diarrhea-associated pathogen was detected in 80.14% (n = 117) of the study population. The most prevalent pathogens were rotavirus 38.36% (n = 56), adenovirus 40/41 19.86% (n = 29), *Shigella/*EIEC 12.33% (n = 18), norovirus GII 11.44% (n = 17) and *Cryptosporidium* 9.59% (n = 14). Co-infections were detected in 26.03% of the study population (n = 38). The presence of multiple pathogens in the stool samples of children with diarrhea indicates poor sanitation and may have significant implications for disease management and patient outcomes.

## 1. Introduction

Diarrhea is among the leading causes of mortality for children, accounting for approximately 9% of all deaths among children aged under five years worldwide. The burden is high among children living in South Asia and Sub-Saharan Africa [1]. Children are more likely to be impacted by diarrheal diseases due to their immature immunity. Diarrhea is caused by various enteric pathogens such as viruses, bacteria, protozoa, helminths and fungi [2,3,4]. These pathogens are commonly spread through consuming contaminated water, food or objects contaminated with fecal matter [5,6].

Different countries report varying proportions of etiology-specific estimates for diarrheal pathogens. However, rotavirus, adenovirus, diarrheagenic *Escherichia coli* (DEC) and *Shigella* remain the most common pathogens causing childhood diarrhea [3,7,8,9,10]. Rotavirus has been the leading global concern driving childhood diarrhea. 

In Tanzania, rotavirus alone accounted for an estimated 8171 out of 11,391 diarrhea cases in children aged under five in 2008 [11]. In response to this, Tanzania introduced the rotavirus vaccine into its national immunization program in January 2013. The vaccine used in Tanzania is the monovalent rotavirus vaccine (RVI), also called Rotarix, which is manufactured by GlaxoSmithKline Vaccines, King of Prussia, PA, USA. RV1 is a live attenuated vaccine given orally in two doses, at 6 and 10 weeks of age, alongside other routine childhood vaccines. According to the Tanzania Demographic and Health Survey 2015–2016, 94% and 89% of children aged 12–23 were reported to have received the first and second doses of Rotarix, respectively. However, this coverage varied across zones. The vaccination coverage for all basic vaccines in the Kilimanjaro region was 93% [12]. Studies conducted in the Dar es Salaam and Manyara regions reported other multiple causative agents of diarrhea in children under five years old [13,14]. However, immunization programs in Tanzania reduced diarrhea cases [15] and hospitalization [16]. 

The presence of multiple enteropathogens can affect the severity of gastrointestinal illnesses [4,17,18]. However, only a few Tanzanian studies have reported co-infections in diarrhea cases, and they differ in the number of pathogens detected. The heterogeneous reports may be due to the design of the research or the techniques used. There are various methods available for detecting and identifying diarrhea-causing pathogens, each with varying cost, sensitivity and specificity [19]. Conventional methods such as culture-, microscopy- and antigen-based assays are commonly used in low-resource settings, but they have limitations, particularly in terms of sensitivity [20]. To overcome these limitations, molecular diagnostic assays offer high sensitivity and the timely detection of pathogens, which is difficult to achieve through conventional methods [19]. Correctly identifying the causative agent of a disease is crucial for effective treatment. However, conventional detection methods are often time-consuming and labor-intensive, and they may not detect slow-growing pathogens such as Campylobacter species, which are important foodborne pathogens globally [21]. Consequently, in low-resource settings, many cases of diarrhea go without a known cause [20]. 

Since the prevention and control of diarrhea are complicated by some pathogens that are not only transmitted through foods but also through contact with animals, diarrheal disease remains the leading cause of mortality and there is a need to investigate a broad spectrum of pathogens in order to guide future interventions. In the present study, customized TaqMan Array card (TAC) was the molecular technique used to detect enteric pathogens from stool samples of children under five years old, who were admitted to health facilities located in Moshi, Kilimanjaro, and to determine the effect of co-infection on the clinical presentation of the children during hospital admission. This study contributes data on etiologies of diarrhea, which in most clinical settings in the study area are unknown. 

## 2. Materials and Methods

### 2.1. Study Design and Settings

This study utilized archived diarrhea samples from the Foodborne Disease Epidemiology, Surveillance and Control in African Low- and Middle-Income Countries (FOCAL) project. Samples were collected between July 2020 and February 2022 from children under five years old who were admitted with diarrhea to 8 health facilities (n = 4 rural; n = 4 urban), which were purposefully selected from healthcare facilities located around the two Moshi Districts in the Kilimanjaro region with the capacity of admitting children and with previous records on the number of diarrhea cases of both inpatients and outpatients. The Moshi Municipal healthcare facilities were Kilimanjaro Christian Medical Center (KCMC), Mawenzi Regional Referral Hospital (MRRH), St. Joseph Hospital and Pasua Health Centre. The Moshi District healthcare facilities were Kibosho Hospital, Kilema Hospital, Marangu Hospital and Umbwe Health Center. Kilimanjaro has a population of around 1.6 million people, including approximately 191,906 children under the age of five [22]. The majority of people in the study areas are engaged in agriculture, primarily growing crops such as maize, banana and beans, with other crops such as rice, cereals, mangoes, groundnuts, sunflowers and tomatoes also being grown. Many households in the area also have livestock and poultry. 

Children aged 0–59 months, admitted due to diarrhea at any of the health facilities participating in this study, were enrolled consecutively. The child should have lived in the same household as the caretaker for at least three months. Children who contracted diarrhea while hospitalized, those exposed to antibiotics upon admission for more than 24 h before enrollment and those with a chronic medical illness such as tuberculosis, kidney disease, liver disease or any type of enteropathy (including cystic fibrosis, Crohn’s disease, celiac disease, ulcerative colitis or malabsorption syndrome) were excluded from the study.

### 2.2. Specimen and Data Collection

One stool specimen was collected from each child in a sterile container within 24 h of admission and transported to Kilimanjaro Clinical Research Institute—Biotechnology Laboratory for analysis. Trained research assistants provided the containers and collection papers, and the child’s caretaker was asked to collect the stool. Samples were transported in a cold chain and stored for further testing. A total of 146 archived diarrhea samples were retrieved and analyzed in this study.

In addition, at the time of admission, the health facility record was consulted to obtain the child’s recorded temperature to determine the presence of a fever (temperature ≥ 38 °C). Furthermore, information pertaining to the child’s demographics, disease history (including any signs and symptoms) and vaccination status were collected from the parents or caretaker through a face to face interview, utilizing a FOCAL project source attribution study questionnaire. 

### 2.3. Laboratory Procedures

Methods used in this study are described in detail elsewhere [23,24]. Briefly, archived samples were removed from the freezer and thawed at room temperature. Total nucleic acid (TNA) was extracted from 180 to 200 mg of stool using QIAamp Fast DNA stool Mini kit (Qiagen, Hilden, Germany). The stool was first lysed by InhibitEx buffer, beaten for 2 min with 212 to 300 µm glass beads, and boiled for 10 min. Other steps were carried out as per the manufacturer’s recommendations. Phocine herpesvirus (PhHV) and MS2 bacteriophage were spiked to the lysis buffer to monitor the efficiency/inhibition of extraction and amplification procedures. The elution volume of TNA was 200 µL and no-template extraction was included in every batch to monitor contamination. The polymerase chain reaction (PCR) mixture was prepared for eight samples and consisted of 425 μL 2× AgPath One Step RT PCR buffer, 34 μL enzyme mix and 234 μL nuclease-free water. An amount of 80 μL of the mastermix was aliquoted into each of 8 Eppendorf tubes, and 20 uL of total nucleic acid was added, mixed gently and centrifuged to remove bubbles from the mixture. Each 100 μL of reaction mix was transferred to the TAC fill port, centrifuged and sealed, and loaded to the ViiA^TM^ 7 Real Time PCR instrument (Applied Biosystems). The details of TAC panel pathogen targets, as well as primer–probe sequences and their respective cut-off points for positivity, are available in the Appendix A. The PCR cycling conditions were reverse transcription (1 cycle) at 45 °C for 20 min, denaturation (1 cycle) at 95 °C for 10 min, PCR (40 cycles) at 95 °C for 15 s, and annealing and extension at 60 °C for 1 min. 

### 2.4. Data Analysis Plan 

Data were analyzed using STATA version 15 (STATA Corp, College Station, TX, USA). Proportions were calculated for the demographic and clinical characteristics of the study participants. The median and interquartile range (IQR) were calculated for the age of the children. The outcome variable was the presence of any enteropathogens (bacteria, viruses, protozoa, helminths or fungi) calculated as a proportion of the samples with positive pathogen targets detected over the total number of samples analyzed. A pathogen was considered as diarrhea-associated if it was detected at a quantity for which the lower 95% CI of the Odds ratio exceeded 1 [22]. We adopted quantification cycles (CTs) cut-off points established based on the pathogen quantities as a criteria to categorize diarrhea-associated pathogens from other positive enteric pathogens which are considered positive at CT ≤ 35 in the TAC assay. Samples with only one positive target were classified as mono-infection and those with more than one positive pathogen target were classified as co-infections. The differences in these two categories were compared by using Chi-squared or Fisher exact tests if the number of observed counts in a cell was below five for both socio-demographic and clinical symptoms. A generalized linear model using the Poisson family was used to calculate an incidence rate ratio (IRR) for the effect of co-infection on clinical symptoms in both univariate and multivariable analysis (Appendix A). The signs and symptoms with a *p*-value of ≤ 0.2 in the univariate analysis were considered in the multivariable model. The final model was obtained by stepwise regression using backward elimination criteria. Factors with a *p*-value < 0.05 were considered statistically significant. 

## 3. Results

A total of 146 children admitted with diarrhea with a median age of 11.74 months (IQR: 7.41–19.09) were included in the analysis. Half of the study participants were aged between 0 and 11 months and 56.85% were from rural areas of the Moshi District. Vomiting (88.36%), fever (60.27%), running nose (45.21%), coughing (45.21%) and loss of appetite (45.89%) were the major clinical manifestations (Table 1). 

### 3.1. Detection of Diarrhea-Associated Pathogens among Study Participants

Based on the previously established cut-off criteria for diarrhea-associated pathogens, at least one enteric pathogen was detected in 80.14% (n = 117) of the study population. The most prevalent diarrhea-associated pathogens were rotavirus 38.36% (n = 56), adenovirus 40/41 19.86% (n = 29), *Shigella/*EIEC 12.33% (n = 18), norovirus GII 11.44% (n = 17) and *Cryptosporidium* 9.59% (n = 14) (Table 2). Out of all the samples analyzed, 29 samples (19.86%) did not test positive for any diarrhea-associated pathogen but did show the presence of other enteric pathogens. Additionally, nine samples (6.16%) did not have any enteric pathogens detected.

### 3.2. Pathogen Detection by Age Categories 

Across all age groups, viruses were the most commonly detected pathogen, followed by bacteria and protozoa. About half of the children in each age category tested positive for the viruses. A high proportion of diarrhea-associated pathogen positivity was observed among children aged between 0 and 11 months. Adenovirus and rotavirus infections were common in all age groups. Infection with *Shigella/*EIEC was higher in children aged 0–11 months and 24–59 months and this difference was significant (Fisher’s exact p = 0.011). Although the proportion of positivity for some enteropathogens, including adenovirus, rotavirus, norovirus GII, *Campylobacter*, EPEC, *Cryptosporidium* and *H. pylori*, were found to decrease with an increase in age, the decrease was not significant (Table 3).

### 3.3. Co-Infections 

Co-infections were detected in 26.03% of the study population (n = 38). Twenty-eight participants (19.18%), nine participants (6.16%) and one participant (0.68) had two, three and four diarrhea-associated pathogens, respectively. Socio-demographic and clinical characteristics of the study participants and their diarrhea pathogen infection status are described in Table 4. Twenty-four (24) different co-infection combinations were detected, and the most common combination was adenovirus + rotavirus infections observed among those with only two pathogens. More details for pathogen combinations are shown in Table 5.

## 4. Discussion

This study shows multiple enteric pathogens detected from children under five years of age admitted with diarrhea to a healthcare facility. The overall results revealed that rotavirus, adenovirus, *Shigella/*EIEC, norovirus GII and *Cryptosporidium* were the most common diarrhea-associated pathogens detected. One quarter of these children were found to have co-infections, which were commonly a combination of viral and bacterial infections. 

In this study, we found that rotavirus was the most prevalent pathogen associated with diarrhea. Rotavirus was detected in children regardless of their vaccination status. Other studies reporting rotavirus as one of the leading causes of diarrhea in under-fives have been conducted elsewhere [8,9,25,26,27,28]. Our results show that there was a decrease in the proportion of rotavirus infections with an increase in age, as is a common trend in children. One possible explanation for this pattern could be the inclusion of a significant proportion of infants aged 0–11 months in the study sample, which may reflect the healthcare-seeking behavior of parents or caregivers who are more likely to seek medical attention for very young children. The prevalence of rotavirus reported in this study is high compared to the prevalence of rotavirus that was reported before [13,14] and after the introduction of the Rotarix vaccine in Tanzania [29,30]. The observed differences in rotavirus prevalence can be explained by variations in the sample sizes between the studies and the technique used for detection of rotavirus. Despite the observed number of diarrhea cases, the vaccine has shown to be effective in reducing the number of diarrhea cases requiring hospitalization in Tanzania [15,31]. Further, a recent study in Kenya found no significant reduction in rotavirus prevalence before and after the introduction of the vaccine [32]. Additionally, adenovirus 40/41 was the second leading viral infection in this study. Similar findings were reported from the Global Enteric Multicenter Study (GEMS) and other studies conducted in other countries [10,24,26,33,34]. The adenovirus 40/41 prevalence reported in this study is higher than the prevalence found in Dar es Salaam [35]. The difference might be due to the different approaches (ELISA versus TAC-PCR) used for the detection of the virus in the samples. 

Other prevalent diarrhea pathogens include the *Shigella/*EIEC species, the leading bacterial pathogen. Species diversity for *Shigella* infection is reported in different geographical areas. For instance, *Shigella/*EIEC was reported to be among the top ten causes of diarrhea in Malawi [8]. In Sudan, *Shigella sonnei* and *Shigella flexeneri* were found to be the more prevalent species [36], indicating species diversity. Our findings reveal a high prevalence of *Shigella/*EIEC compared to findings in Ethiopia and Saudi Arabia [25,37,38]. By contrast, our findings reveal a lower prevalence of *Shigella/*EIEC compared to a study carried out in Niger [39]. *Shigella* infection results in moderate–severe diarrhea and it has been reported to be associated with the largest number of diarrhea cases resulting from the consumption of contaminated food [40]. The 2016 report on the Global Burden of Disease reported *Shigella* to be the second leading cause of diarrhea-related mortality [41]. Mortality related to *Shigella* is also reported in GEMS analysis [42]. Although some studies reported the presence of the *Shigella* species to be more common in bloody diarrhea [38,39], our study did not find any bloody diarrhea cases.

Our study found co-infections in more than one quarter of the study participants. More than three quarters of the co-infections were observed in children below two years of age. This may have been influenced by poor environmental sanitation and children’s hand-to-mouth habits. We found diverse combinations of co-infections. However, rotavirus–adenovirus and rotavirus–*H. pylori* combinations were detected in at least five cases. The fact that the detected co-infections had combinations of pathogen groups (viruses, bacteria and protozoa) might have implications for diarrhea treatment, which is empirically based in most developing countries. Another study reporting co-infection was conducted in Dar es Salaam [43], which also found co-infection in a large proportion of cases and controls. The simultaneous detection of multiple pathogens has been associated with an increase in the severity of the clinical symptoms [18] and pathogenicity as compared to mono-infection [17,43]. However, this was not detected in this study.

Given that this study was conducted during COVID-19, there is limited data on the effect of COVID-19 on diarrheal incidence in under-five children in Tanzania. However, studies conducted in other countries have suggested that the COVID-19 pandemic-related changes in hygiene practices and health-seeking behaviors may have affected the incidence of diarrhea diseases in children. One study in the Democratic Republic of Congo (DRC) found a significant decrease in the number of admissions due to diarrhea among under-five children during the COVID-19 pandemic period compared to previous years [44]. Another study conducted in Ethiopia reported a slight decrease in the number of diarrheal cases in under-fives before and during the COVID-19 pandemic; however, the decrease was not statistically significant [45]. The variations between countries might be due to the different implementation and effectiveness of public health interventions, or to differences in cultural and socio-economic factors. 

This study has analyzed co-infections in the study area which can positively influence the diagnosis of childhood diarrhea. The major advantage of multipathogen detection in diarrhea cases is that it helps to guide treatment decisions, which improves patient outcomes. Diarrheagenic *E. coli* can be hard to distinguish from *E. coli* that is part of normal flora in the human gut. Our study enabled a number of diarrhea cases to be identified as DEC that would otherwise have been reported as an unknown etiology due to the lack of resources. Further, our study found no enteric pathogens in only nine samples (6.16%). This could be explained by the following reasons: not all diarrhea is infectious, diarrhea can be post-infectious and the TAC panel does not cover all potential enteric pathogens. Moreover, our study has some limitations, including the small sample size used and sampling bias as a result of inconsistency observed during sampling in the parent study that resulted in samples having less than the required volume.

## 5. Conclusions

In this study, multiple pathogens were detected in the stool samples of children under five years of age with diarrhea. This study might have an impact on disease management. Although these findings are not generalizable, utilizing more sensitive techniques could aid in identifying disease-causing agents and generating surveillance data that could improve future disease treatment plans. In addition, promoting good hand hygiene, ensuring access to safe drinking water and sanitation, and encouraging immunization continue to be crucial interventions in this regard. The occurrence of multiple pathogens in the gut of the children might be the result of a poor sanitary environment that harbors different microorganisms. In such an environment, pathogens pass to children through poor hygiene during food preparation or by the children drinking contaminated water.

## Figures and Tables

**Table 1 pathogens-12-00618-t001:** Demographic and clinical characteristics of the study participants (N = 146).

Characteristic	Frequency	Percentage
**Child Sex**		
Female	72	49.32
Male	74	50.68
**Age**		
0–11 months	74	50.68
12–23 months	44	30.14
24–59 months	28	19.82
**Area of Residence**		
Rural	83	56.85
Urban	63	43.15
Rotavirus vaccination status *	89	67.42
**Clinical signs and Symptoms**		
Max. stool frequency/24 h		
3–5	82	56.16
≥6	64	43.84
Vomiting	129	88.36
Fever	88	60.27
Loss of appetite	67	45.89
Coughing	66	45.21
Running nose	66	45.21
Fatigue	12	8.22
Abdominal Pain	6	4.11
Nausea	6	4.11
Flatulence	5	3.42
Headache	1	0.68
Jaundice	1	0.68

* Rotavirus vaccination status was recorded only for children with clinic cards on the day of enrollment.

**Table 2 pathogens-12-00618-t002:** Number of diarrheal pathogens detected in the stool samples of the study participants (N = 146).

Pathogen	n (%)
Adenovirus	29 (19.86)
Astrovirus	1 (0.68)
*C. jejuni/coli*	5 (3.42)
*Cryptosporidium*	14 (9.59)
EPEC	8 (5.48)
ETEC	5 (3.42)
*H. pylori*	7 (4.79)
Norovirus GII	17 (11.64)
Rotavirus	56 (38.36)
*S. enterica*	1 (0.68)
Sapovirus	4 (2.74)
*Shigella*/EIEC	18 (12.33)
*V. cholerae*	1 (0.68)

**Table 3 pathogens-12-00618-t003:** Distribution of diarrhea-associated pathogens detected by age categories.

Pathogen	Positive Samples	0–11 Months (N = 74)	12–23 Months N = 44	24–59 Months N = 28
Pathogen by group	n(%)	n(%)	n(%)	n(%)
Viruses	91	43 (58.11)	33 (75.00)	15 (53.57)
Bacteria	40	19 (25.68)	10 (22.73)	11 (39.29)
Protozoa	14	7 (9.46)	5 (11.36)	2 (7.14)
Specific pathogens				
Adenovirus	29	14 (18.92)	10 (22.73)	5 (17.86)
Rotavirus	56	23 (31.08)	21 (47.73)	12 (42.86)
Norovirus_GII	17	10 (13.51)	6 (13.64)	1 (3.57)
Sapovirus	4	1 (1.35)	3 (6.82)	-
Astrovirus	1	1 (1.35)	-	-
*C. jejuni*	5	3 (4.05)	2 (4.55)	-
EPEC	8	5 (6.76)	2 (4.55)	1 (3.57)
ETEC	5	2 (2.70)	2 (4.55)	1 (3.57)
*H. pylori*	7	4 (5.41)	2 (4.55)	1 (3.57)
*S. enterica*	1	1 (1.35)	-	-
*Shigella/*EIEC	18	8 (10.81)	2 (4.55)	8 (28.57)
*V. cholerae*	1	-	1 (2.27)	-
*Cryptosporidium*	14	7 (9.46)	5 (11.36)	2 (7.14)

**Table 4 pathogens-12-00618-t004:** Socio-demographic and clinical characteristics of the study participants by diarrhea-associated pathogen detection and infection status (N = 146).

Characteristic	No Infection (N = 29) n (%)	Mono-Infection (N = 79) n (%)	Co-Infection (N = 38) n (%)	Total n (%)
Child Sex				
Female	20 (68.97)	40 (50.63)	12 (31.58)	72 (49.32)
Male	9 (31.03)	39 (49.37)	26 (68.42)	74 (50.68)
Age				
0–11 months	20 (68.97)	35 (44.30)	19 (50.00)	74 (50.68)
12–23 months	6 (20.69)	25 (31.65)	13 (34.21)	44 (30.14)
24–59 months	3 (10.34)	19 (24.05)	6 (15.79)	28 (19.82)
Area of Residence				
Rural	15 (51.72)	47 (59.49)	21 (55.26)	83 (56.85)
Urban	14 (48.28)	32 (40.51)	17 (44.74)	63 (43.15)
Use diaper	3 (10.34)	19 (24.36)	17 (44.74)	39 (26.90)
Max.stool frequency/24 h				
3–5	20 (68.97)	42 (53.16)	20 (52.63)	82 (56.16)
≥6	9 (31.03)	37 (46.84)	18 (47.37)	64 (43.84)
Vomiting	23 (79.31)	72 (91.14)	34 (89.47)	129 (88.36)
Fever	17 (58.62)	48 (60.76)	23 (60.53)	88 (60.27)
Loss of appetite	11 (37.93)	34 (43.04)	22 (57.89)	67 (45.89)
Coughing	16 (55.17)	36 (45.57)	14 (36.84)	66 (45.21)
Running nose	16 (55.17)	38 (48.10)	12 (31.58)	66 (45.21)
Fatigue	1 (83.45)	6 (7.59)	5 (113.16)	12 (8.22)
Abdominal Pain	2 (6.90)	3 (3.80)	1 (2.63)	6 (4.11)
Nausea	-	4 (5.06)	2 (5.26)	6 (4.11)
Flatulence	-	2 (2.53)	3 (7.89)	5 (3.42)
Headache	-	1(1.27)	-	1 (0.68)
Jaundice	1 (3.45)	-	-	1 (0.68)

**Table 5 pathogens-12-00618-t005:** Co-infection combinations detected among diarrhea cases.

Co-Infections for Diarrhea Associated Pathogens	Frequency
*Cryptosporidium* + rotavirus	1
*Cryptosporidium* + adenovirus	2
*Cryptosporidium* + *Shigella/*EIEC	1
*H. pylori* + rotavirus	3
*H. pylori* + **adenovirus**	1
Norovirus + adenovirus	2
Sapovirus + *Shigella/*EIEC	1
Rotavirus + adenovirus	7
*C. jejuni* + rotavirus	1
*Cryptosporidium* + rotavirus	2
ETEC + rotavirus	3
ETEC + norovirus	1
EPEC + rotavirus	1
EPEC + *Shigella/*EIEC	1
EPEC + astrovirus	1
Adenovirus + *Shigella/*EIEC *+ C. jejuni*	1
Adenovirus + rotavirus *+ Shigella/*EIEC	1
Adenovirus + rotavirus *+* ETEC	1
Adenovirus + rotavirus *+* sapovirus	1
Adenovirus + *C. jejuni + Shigella/*EIEC	1
Adenovirus + *H. pylori +* norovirus	1
Adenovirus + *C. jejuni + V. cholerae*	1
*H. pylori* + rotavirus *+* sapovirus	1
*H. pylori* + rotavirus *+ Shigella/*EIEC	1
Adenovirus + Norovirus *+ C. jejuni* + *Cryptosporidium*	1

## Data Availability

The data presented in this study are available on request from the corresponding author.

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
