# Peer review of "Enteric Pathogens Detected in Children under Five Years Old Admitted with Diarrhea in Moshi, Kilimanjaro, Tanzania"

_pathogens, 2023, doi:10.3390/pathogens12040618_

Round 1

Reviewer 1 Report

In this manuscript, Hugho et al. report on detection of enteric pathogens in children <5 years of age with diarrhea in the rural region of Moshi-Kilimanjaro, Tanzania. They leverage an archived specimen bank to perform nucleic acid extraction and pathogen detection using Taqman array card. Results were analyzed for pathogen detection, co-infections, and clinical presentation. They found that rotavirus and adenovirus 40/41 were the most prevalent pathogens associated with diarrhea in hospitalized children, and three of the top five were viruses (norovirus G.II being the third). Over a quarter of children had enteric coinfection, but this was not associated with any meaningful clinical outcomes. Strengths of the study are the well-characterized cohort and use of TAC for comprehensive enteropathogen detection. Limitations include small sample size and concerns about generalizability. While not novel, the data nevertheless provide additional supportive evidence of the importance of viral pathogens in pediatric diarrhea in low-income settings, particularly the ongoing challenge of rotavirus despite vaccine introduction. I believe the overall quality of the writing can be improved and there are specific aspects of the manuscript, as detailed below, that would require revision

Abstract

·       Be consistent with use of “coinfection” vs “co-infection.”

·       Would suggest more general keywords (e.g. diarrhea, diarrheal pathogens, children, etc.)

Introduction

·       Much of the general background info presented here seems extraneous and could be omitted or more concisely summarized, for example lines 36-42, 53-58, and 60-65.

·       More general background information on the rotavirus immunization program (vaccine used, introduction, coverage, etc.) in Tanzania would be helpful

·       Line 50: what is the difference between symptoms and disease conditions?

·       Lines 66-69: are emerging diarrheal pathogens really the main consideration here?

Materials and Methods

·       Lines 78-87 seem unnecessary, as the more pertinent information is found in the paragraph below. The overall agricultural importance of the region could perhaps be conveyed in a single sentence after the description of the study sites ending in line 98.

·       There is no information regarding ethical approval or informed consent.

·       Lines 108-118 are overly descriptive and should be shortened.

·       Would consider including the TAC targets as a table rather than text, and include the reference demonstrating the PCR primers and probes used

·       Line 143: the annealing step and extension steps are missing from the PCR description. I assume the extension at 60C refers to the final extension, not the cycling step?

·       Were the Ct cut-offs the same for each pathogen? If not, I would include in the table of targets.

·       How was incidence calculated? Were regional census data available to calculate the total population of children at risk during the study period?

·       The generalized linear model requires more detailed explanation. What was the outcome assessed? What were the predictors analyzed? What was included in the multivariable analysis, and based on what selection criteria?

Results

·       Is rotavirus vaccination status available for each child? This would be of critical importance.

·       I would consider relabeling table 1 as “Clinical signs and symptoms.” Fever and jaundice are both signs. Also, how was fever defined? Were temperature criteria used or was parent report of concern for fever accepted? Loss of appetite, fatigue, abdominal pain, nausea, and headache cannot be reported by infants and young children. How were these data acquired? If they were only positively reported by older children, this would significantly bias these results.

·       Lines 174-175: what does this mean?

·       Lines 175-176: same question

·       Table 2: what is the significance of the bolding? Why are some genus names spelled out and not abbreviated, when all were already fully spelled out earlier?

·       Lines 181-2 could be better worded

·       Lines 185-187: suggest wording to state that infection was significantly higher in children 24-59 months, although I question the clinical significance, given the limited sample size

·       Line 189: can delete “an increase in”

·       Again, consistency in use of “coinfection” vs “co-infection” is needed

·       Line 192: suggest deleting “For analysis of diarrhea-associated pathogens” and “(2 to 4)”

·       Table 4: would list total N in column headings, and then can remove “No” rows for each category to just show the positives

·       Table 5: again, what is the significance of the bolding? And what is the significance of the combinations for highly diarrhea associated pathogens at the end (i.e. what does this mean?)

·       Table 6: again, suggest total N in column headings. I am confused by what this is showing. What are the multivariable results? Why are some shown with an adjusted IRR and some not? How was the adjustment performed?

·       The overall findings from Table 6 and associated text are of questionable clinical significance, I would consider moving the table to a supplemental table.

Discussion

·       Line 229-230 seems out of place

·       Lines 213-232 are unclear

·       There is no discussion on whether or not these rotavirus infections were breakthrough infections in vaccinated children or not

·       Lines 260-263 are unclear and could be improved

·       Again, I think the clinical findings presented here for coinfection are of dubious significance

·       Low sample size is another significant limitation that is not mentioned

·       Because TAC would not be widely available for routine clinical diagnostics, the argument that these results would help improve treatment decisions are not justified, unless this logic can be better explained

·       There should be some discussion of the fact that the study period was during COVID-19, and how that may have affected overall diarrhea incidence during the study period

·       Of the pathogens assessed, only rotavirus has an available vaccine and yet it remained the most prevalent cause of diarrhea. In this context, I find line 291 a little difficult to interpret. Do the authors mean that better vaccines need to be developed? I fully support rotavirus vaccination, but there is no vaccine data here to support the notion that it would be protective in this population.

Reviewer 2 Report

This manuscript by Ephrasia et al. investigated diarrhea-causing pathogens and determined the effect of co-infection on clinical symptoms. The results display that the most prevalent pathogens were Rotavirus and Adenovirus, and co-infections were detected in 26.03%. The results may have significant implications on disease management and patient outcomes. This manuscript is scientifically sound; however, minor revisions are required as follows.

Materials and Methods

2.3. Laboratory procedures.

Did the author conduct multiple RT-PCR tests on multiple pathogens? Or for single pathogen?

Primers for different pathogens need to be provided in table, because these are key test information.

Round 2

Reviewer 1 Report

The authors have thoughtfully responded to the previous comments raised by this reviewer. A number of additional minor edits (the majority to improve readability) are suggested, as detailed below.

Introduction

·       Line 38 typo: “helminthes”

·       Line 45: no need to capitalize Rotavirus

·       Line 47: “its”

·       Line 49 typo: GlaxoSmithKline

·       Line 53: “This coverage”

·       Line 66: no need to capitalize Molecular

·       Lines 73-74: not sure how contact with animals is relevant here in the context of this paper

·       Line 75: delete “and”?

Materials and Methods

·       Line 98: “households”

·       Line 139: ViiA7 manufacturer?

·       Line 140: “are found in are available in” needs to be corrected

·       Supplemental Table 1 was not included in the revised submission

Results

·       Suggest adding the rotavirus vaccination status of each child into Table 1

·       Line 170: “Diarrhea” does not need to be capitalized and should read “diarrhea-associated”

·       Lines 175-176: I’m still not sure what the difference is between a diarrhea-associated pathogen and an enteric pathogen.

·       Table 2 has is not referenced in the text anywhere.

·       Table 2: Would suggest removed “Diarrhea associated quantities” from column header and simply say N (%), and change title to “Number of diarrhea pathogens detected in the stool samples of study participants at diarrhea-associated quantities”

·       Lines 173, 185, 189, 198: No need to capitalize the viruses

·       Table 3, headers need to specify numbers reported are n (%)

·       Table 4: The Yes lines can be removed to just report the number of children with each symptom. E.g.: Vomiting        23 (79.31)           72 (91.14) etc.

As currently written, the table is very confusing to read.

Discussion

·       As elsewhere, viruses do not typically require capitalization

·       Line 214: consider rephrasing, “Rotavirus was frequently detected in previously vaccinated children.”

·       Lines 216-220: I believe this simply reflects the natural epidemiology of rotavirus, which has peak incidence in most low-income settings during infancy

·       Lines 221-223: Unclear

·       Lines 224-226: I’m not clear what differences are being referred to, and how the study era would explain this since both the cited Kenya and Tanzania studies include pre- and post-vaccine time periods.

·       Line 242: “Global Burden of Disease”

·       Line 250: “at least”

·       Line 251: suggest “The fact that detected co-infections frequently had combinations of pathogen groups (viruses, bacteria, protozoa) might have implications for diarrhea treatment, which is empirically based in most developing countries.”

·       Line 256-7: would specific that this was not detected in this study however

·       Line 265: “reported a slight decrease in”

·       Line 268: suggest adding “or” after the comma

·       Lines 270-273: wording is confusing

·       Line 274: “from E. coli” is repeated

Author Response

Dear Reviewer,

Thank you for your comments and suggestions.

Please find the attachment with the responses.

Regards,

Ephrasia
